# Efficacy and Safety of the Topotecan–Cyclophosphamide Regimen in Adult Metastatic Ewing Sarcoma: A Large, Multicenter, Real-World Study

**DOI:** 10.3390/cancers17030550

**Published:** 2025-02-06

**Authors:** Salih Tunbekici, Haydar Cagatay Yuksel, Caner Acar, Gokhan Sahin, Oguzcan Kınıkoglu, Nargiz Majidova, Mustafa Alperen Tunç, Mürsel Sali, Adem Deligonul, Berkan Karadurmus, Ibrahim Tunbekici, Pınar Gursoy, Ulus Ali Sanli, Erdem Goker

**Affiliations:** 1Department of Medical Oncology, Faculty of Medicine, Ege University, 35100 Izmir, Turkey; haydar.cagatay.yuksel@ege.edu.tr (H.C.Y.); caner.acar@ege.edu.tr (C.A.); gokhan.sahin@ege.edu.tr (G.S.); pinar.gursoy@ege.edu.tr (P.G.); sanliua@gmail.com (U.A.S.); erdem.goker@ege.edu.tr (E.G.); 2Department of Medical Oncology, Kartal Dr. Lütfi Kirdar City Hospital, Health Science University, 34865 Istanbul, Turkey; ogokinikoglu@yaho.com; 3Department of Medical Oncology, School of Medicine, Marmara University, 34899 Istanbul, Turkey; nargiz.majidova@medicalpark.com.tr (N.M.); m.alperen.tunc@gmail.com (M.A.T.); 4Department of Medical Oncology, Uludag University, 16059 Bursa, Turkey; murselsali@uludag.edu.tr (M.S.); ademd@uludag.edu.tr (A.D.); 5Department of Medical Oncology, Gülhane Faculty of Medicine, University of Health Sciences, 06018 Ankara, Turkey; berkankaradurmus@gmail.com; 6Department of Plastic Surgery, Cukurova University, 01330 Adana, Turkey; i.tunbekici2@gmail.com

**Keywords:** Ewing sarcoma, second-line chemotherapy, topotecan, cyclophosphamide, efficacy, safety, metastasis

## Abstract

Ewing sarcoma is a malignant tumor that frequently occurs in adolescents and young adults, primarily originating from bone and rarely from soft tissue. Approximately 25% of patients present with metastatic disease at the time of diagnosis. Ewing sarcoma is considered a systemic disease, with the standard treatment consisting of a multimodal approach that combines surgery, radiotherapy, and chemotherapy. However, the optimal second-line or subsequent-line treatment for patients who progress after first-line chemotherapy remains unclear. Treatment options include high-dose ifosfamide, gemcitabine–docetaxel, topotecan–cyclophosphamide, and vincristine–irinotecan–temozolomide. The aim of this study was to evaluate the efficacy and safety of the topotecan–cyclophosphamide regimen in adult patients with metastatic Ewing sarcoma. The topotecan–cyclophosphamide regimen appears to be a viable treatment option regarding efficacy and safety for patients who progress after first-line therapy. However, further research is needed to better define its role in treatment.

## 1. Introduction

Ewing sarcoma (ES) is a malignant tumor that frequently occurs in adolescents and young adults, primarily originating from bone and rarely from soft tissue [1,2]. Although ES typically occurs in the long bones of the extremities and the flat bones of the pelvis, it can be observed in all types of bones [3]. There are significant clinical differences between skeletal and extraskeletal ES. Patients with extraskeletal ES, in contrast to skeletal ES, are typically diagnosed at an older age and are often female [4]. ES usually presents with localized pain or swelling. Approximately 10–20% of patients may also exhibit systemic symptoms, such as fever, fatigue, weight loss, and anemia [5,6,7]. Clinical and biological characteristics, such as the presence of metastases, the primary tumor’s location and size, patient age, and response to treatment, are critical determinants of disease prognosis [8,9,10,11]. Local treatments are often insufficient for disease control; therefore, multimodal treatment approaches, including chemotherapy, radiotherapy, and surgical intervention, are considered the standard of care [12]. Approximately 75% of patients present with localized disease, with a 5-year survival rate of about 70%. In contrast, patients with metastatic disease at diagnosis have a 5-year survival rate of 20% [13].

Small blue round-cell sarcomas (SBRCSs) are a heterogeneous group of mesodermal neoplasms. According to the current World Health Organization classification of soft tissue tumors, SBRCSs are composed of relatively undifferentiated cells with small blue nuclei and a round to oval cytomorphology in hematoxylin and eosin staining. Overlapping histological and morphological features make the routine morphology-based classification of SBRCSs into specific subgroups challenging. Advances in cytogenetics and molecular genetics have facilitated the detection of tumor-specific alterations in certain subgroups of SBRCSs. These alterations are characterized by particular recurrent chromosomal rearrangements and associated specific gene fusions. ES is the most common type of SBRCS. Genetically, specific chromosomal translocations are commonly observed in ES patients. The most frequent of these is the EWSR1-FLI1 translocation (85%). In cases where this translocation is absent, the next most common translocation is EWSR1-ERG [14,15,16,17,18].

Due to the exclusion of adults from most studies, there are few randomized trials on the treatment of metastatic ES in adult patients. Therefore, the treatment approach is similar to that used for children [19]. In the first-line treatment of metastatic ES, the addition of ifosfamide and etoposide (IE) to a vincristine, doxorubicin, and cyclophosphamide (VDC) regimen is considered standard treatment [20]. There are no standard treatment options for second-line and subsequent lines of therapy. Guidelines recommend the following treatment options: high-dose ifosfamide (HDIFO), which is administered at 3 g/m^2^ on days 1–5 every 21 days, along with mesna and hydration support; irinotecan 50 mg/m^2^ and temozolomide 100 mg/m^2^ (IT), which are administered on days 1–5 every 21 days; vincristine 1.5 mg/m^2^ on day 1, irinotecan 50 mg/m^2^ on days 1–5, and temozolomide 100 mg/m^2^ on days 1–5 (VIT), which are administered every 21 days; topotecan 0.75 mg/m^2^ and cyclophosphamide 250 mg/m^2^ (TC), which are administered on days 1–5 every 21 days, along with mesna and hydration support; ifosfamide 1800 mg/m^2^ on days 1–5, carboplatin 400 mg/m^2^ on days 1–2, and etoposide 100 mg/m^2^ on days 1–5 (ICE), which are administered every 21 days; and gemcitabine 900 mg/m^2^ on days 1 and 8, along with docetaxel 80 mg/m^2^ on day 1, which are administered every 21 days (GD), for patients who progress after receiving the IE/VDC regimen [21]. Only a few studies in the literature have investigated the efficacy and safety of the TC regimen in adult patients with metastatic ES. The goal of this multicenter, real-life study was to evaluate the efficacy and safety of the TC regimen in adult patients with metastatic ES who had previously been treated with chemotherapy.

## 2. Materials and Methods

### 2.1. Patients and Data Collection

Our study was a retrospective analysis. The study included patients who were followed at five tertiary centers in Turkey between 2011 and 2020. Patients’ data were obtained from patient files. The key inclusion criteria were histologically confirmed metastatic ES, age > 18 years, patients having experienced disease progression after first-line treatment and subsequently receiving the TC regimen, an Eastern Cooperative Oncology Group performance status score of ≤2, and adequate bone marrow, liver, and renal function. Patients under 18 years old, those with localized ES, and patients with missing data were excluded from the study. The TC regimen consisted of topotecan administered at 0.75 mg/m^2^/day and cyclophosphamide at 250 mg/m^2^/day, given daily for 5 days and repeated every 21 days. Each cycle included granulocyte colony-stimulating factor support, and mesna, along with hydration, was used for uroprotection during and after cyclophosphamide infusion. Treatment responses were assessed radiologically every four cycles. Treatment responses were assessed according to the Response Evaluation Criteria In Solid Tumors (RECIST) 1.1 criteria. Treatment-related toxicities were evaluated during each treatment cycle. Adverse effects were graded according to the Common Terminology Criteria for Adverse Events, version 5.

### 2.2. Statistical Analysis

Categorical variables are presented as frequency (n) and percentage (%), and numerical data are shown as the median and range to represent the distribution. Progression-free survival (PFS) was defined as the time from the start of the TC regimen to the date of progression or death. Overall survival (OS) was defined as the time from initiation of the TC regimen to death. The objective response rate (ORR) was defined as the percentage of patients with a complete response (CR) + partial response (PR) according to the RECIST 1.1 criteria. The disease control rate (DCR) was defined as the percentage of patients with a CR + PR + stable disease (SD). The OS and PFS were estimated using the Kaplan–Meier method. Univariate Cox proportional hazards regression analysis was performed to evaluate the association of baseline characteristics with overall survival. The hazard ratio and 95% confidence interval were calculated for each variable. The results were visualized using a forest plot to illustrate the hazard ratios and their confidence intervals. Variables with *p* < 0.05 were considered statistically significant. All analyses were performed using IBM SPSS Statistics software (version 26; IBM Corp., Armonk, NY, USA).

## 3. Results

### 3.1. Patient and Treatment Characteristics

Seventy-five patients were enrolled in the study. The median age was 23 years (range: 18–52). Fifty (66.6%) patients were male, and twenty-five (33.3%) patients were female. At diagnosis, 40 (53.3%) patients presented with localized disease and 35 (46.6%) had de novo metastatic disease. The most common sites of metastasis were the lungs (49.3%) and bones (45.3%). A total of 38 (50.6%) patients received the TC regimen as the second-line treatment, whereas 22 (29.3%) patients received it as the third-line treatment, and 15 (20%) patients received it in the fourth and subsequent lines (patient characteristics are summarized in Table 1).

### 3.2. Survival Outcomes

At the time of the analysis, all 75 (100%) patients had died. The median PFS (mPFS) with the TC regimen was 3.06 months (95% CI, 2.91–3.22) (see Figure 1). The median OS (mOS) with the TC regimen was 6.16 months (95% CI, 5.14–7.18) (see Figure 2). Patients who received the TC regimen in the second line demonstrated longer OS (7.55 months 95% CI, 5.37–14.17) compared to those who received it in the third line or later (5.70 months 95% CI, 4.07–6.60) (*p* = 0.005). Patients who were metastatic at the time of diagnosis tended to have a shorter OS compared with those who had localized disease at diagnosis (5.40 months vs. 6.53 months) (*p* = 0.053) (see Table 2) (see Figure 3). Other clinical and treatment-related parameters affecting OS were evaluated, and no statistically significant factor was found.

### 3.3. Response to Topotecan–Cyclophosphamide Regimen

Thirty-eight patients received the TC regimen as second-line treatment, with an ORR of 5/38 (13.2%) and a DCR of 19/38 (50%). Thirty-seven patients received the TC regimen in the third line or later, with an ORR of 3/37 (8.1%) and a DCR of 4/37 (10.8%). In the entire group, the ORR was 8/75 (10.7%) and the DCR was 23/75 (30.7%). The treatment response could not be evaluated in 13 (17.3%) patients; 10 of these patients died, and 3 patients discontinued the TC regimen due to adverse effects (see Table 3).

### 3.4. Safety

Toxicity information was obtained from medical records. Complete reports were available for 68 patients. The median number of cycles was four (range: 1–12). Hematologic adverse effects were observed in 55 (80.8%) patients, with grade 3–4 hematologic toxicities observed in 35 (51.4%). Twenty (29.4%) patients required erythrocyte transfusions, and ten (14.7%) required platelet transfusions. Although the TC regimen caused severe myelosuppression, infectious complications were rare. Grade 3–4 neutropenia was observed in 25 (36.7%) patients, whereas febrile neutropenia was observed in only 5 (7.3%). Three (4.4%) patients were treated with intravenous antibiotics, and two (2.9%) received oral antibiotics. Ten (14.7%) patients required a dose reduction owing to drug-related adverse events. Three (4.4%) patients permanently discontinued treatment because of drug-related side effects (one because of thrombocytopenia, one because of anemia, and one because of interstitial lung disease). Non-hematologic adverse effects included hemorrhagic cystitis, nausea, vomiting, diarrhea, and perirectal mucositis, all of which were manageable. No deaths were considered drug-related (see Table 4).

## 4. Discussion

To the best of our knowledge, this is the largest real-world study investigating the efficacy and safety of the TC regimen in adult patients with metastatic ES. No standard treatment options exist for second-line and subsequent lines of therapy. In these patients, treatment regimens such as IT, VIT, HDIFO, TC, ICE, and GD chemotherapy are considered alternatives, with guidelines prioritizing TC and VIT as the preferred options for second-line treatment [21].

In a study evaluating the efficacy and safety of the VIT chemotherapy regimen in 16 adult patients with metastatic ES, the median age was 25 years. All patients had received at least two lines of chemotherapy, with the most common metastatic site being the lungs. The mPFS was 3.6 months, and the mOS was 5.6 months. PR were observed in 5 (31%) patients, but no complete responses were seen, resulting in an ORR of 31%. Grade 1–2 adverse events were observed in 13 (81.3%) patients, while 5 (31.3%) patients experienced grade 3–4 adverse events. The most commonly observed adverse effects were hematological [22]. In a phase 2 study comparing short- and long-term VIT chemotherapy regimens in 46 patients with relapsed or refractory ES, the short-term regimen consisted of intravenous irinotecan 50 mg/m^2^/day on days 1–5 and vincristine 1.4 mg/m^2^/day (maximum 2 mg) on day 1. The long-term regimen consisted of irinotecan 20 mg/m^2^/day on days 1–5 and 8–12 and vincristine 1.4 mg/m^2^/day (maximum 2 mg) on days 1 and 8. In both regimens, temozolomide (100 mg/m^2^/day) was administered intravenously on days 1–5, one hour before irinotecan. The primary endpoint of the study was the ORR at week 12, while secondary endpoints included PFS and OS. At week 12, the ORR was 20.8% in the short-term regimen and 54.5% in the long-term regimen. The median PFS was 2.3 months for the short-term regimen and 4.3 months for the long-term regimen. The median OS was 14.8 months for the short-term regimen and 12.8 months for the long-term regimen. No statistically significant differences were observed in PFS and OS, but more grade 3–4 adverse events were observed in the short-term regimen. The long-term regimen showed better efficacy and tolerability [23].

rEECur is a multi-arm, multi-stage, randomized phase II/phase III, open-label, international study. Upon entry, patients will be randomized to receive one of the four treatment regimens consisting of IT, GD, TC, and HDIFO. The phase II part consists of two stages. The first stage includes four chemotherapy arms. Once 50 patients have been recruited to each arm, one arm will be dropped based on activity and/or toxicity. The second stage will involve a three-way randomization between the remaining arms. After 25 additional patients have been recruited to each arm, a second arm will be dropped based on activity and/or toxicity. The remaining two arms will progress to phase III evaluation. Patients in these two arms who participated in the phase II stage will contribute data to the phase III stage. In the study design, during phase II, the GD arm was dropped first, followed by the IT arm, while the TC and HDIFO arms progressed to phase III. In the phase III comparison between TC and HDIFO (both 73 patients), the median Event-Free Survival was 3.7 months (95% CI, 2.1–6.2) for TC and 5.7 months (95% CI, 3.8–7.0) for HDIFO. The median OS was 10.4 months (95% CI, 7.5–15.5) for TC and 16.8 months (95% CI, 11.1–25.8) for HDIFO. A significant survival difference was observed for both Event-Free Survival and OS between patients under 14 years of age and those aged 14 years and older. It was found that TC was more toxic than HDIFO in the study [24].

In a pediatric cohort consisting of patients with relapsed and refractory sarcomas, the efficacy and safety of the ICE chemotherapy regimen were investigated. Among the 97 patients, 21 had ES. The ORR was found to be 51%. The mOS was 11.8 months (95% CI: 8.3–17.6). Grade 3–4 hematologic adverse events were observed in all patients [25].

The efficacy and safety of the GD combination were investigated in a study involving 22 patients with refractory bone and soft tissue sarcomas. Seventeen patients had osteosarcoma, two had ES, one had malignant fibrous histiocytoma, one had chondrosarcoma, and one had undifferentiated sarcoma. Fourteen patients were assessable for response. The ORR was 29%. Toxicity was manageable and primarily consisted of thrombocytopenia and neutropenia [26].

In a study examining the effect of high-dose chemotherapy and stem cell transplantation on OS in non-metastatic ES patients, those who responded well to induction therapy continued with maintenance treatment, while patients who showed a poor response to induction therapy received high-dose chemotherapy in addition to conventional chemotherapy, followed by autologous stem cell transplantation. The findings demonstrated that high-dose chemotherapy and stem cell transplantation were both effective and feasible for patients with a poor response to induction therapy [27]. However, in a study examining the effect of high-dose treosulfan and melphalan in induction therapy followed by autologous stem cell transplantation in patients with metastatic ES, the contribution of high-dose chemotherapy and stem cell transplantation could not be demonstrated [28].

Tyrosine kinase inhibitors are also among the treatment options for patients with relapsed, refractory, or metastatic ES. In a phase 2 single-arm study of cabozantinib in advanced ES, the ORR was 26%, and the mPFS was 5 months. Another non-randomized phase 2 study reported a 10% ORR with regorafenib in recurrent ES [29,30].

Gangliosides are molecules composed of glycosphingolipids associated with one or more sialic acid residues. Disialoganglioside GD2, a glycosphingolipid that plays a role in cell proliferation, differentiation, survival, and tumor progression, is a tumor-associated antigen expressed in various malignancies, including neuroblastoma, but is found at limited levels in normal tissues. Immunotherapy using GD2-targeting antibodies is the standard first-line treatment for high-risk neuroblastoma. GD2 expression is also observed in other pediatric solid tumors, such as ES, osteosarcoma, and others. In ES, GD2 expression levels range from 40% to 90% in biopsy samples, making it a potentially valuable therapeutic target in this malignancy. The addition of the anti-GD2 antibody dinutuximab beta to standard chemotherapy regimens has been evaluated for its efficacy in three patients with metastatic, GD2-positive ES or Ewing-like sarcoma. The treatment was well-tolerated, and all patients achieved complete remission without evidence of relapse. First-line anti-GD2 immunotherapy in patients with metastatic, GD2-positive ES or Ewing-like sarcoma represents a promising therapeutic option that warrants further clinical evaluation [31,32].

In a study investigating the efficacy of the TC regimen in relapsed ES in a pediatric population of 16 patients, the ORR was 23%, the SD rate was 31%, and the DCR was 54% [33]. In another study evaluating the efficacy of the TC regimen in relapsed bone sarcomas, 15 patients were included, with a median age of 31 years. Of these patients, six had non-pleomorphic rhabdomyosarcoma, five had ES, two had synovial sarcoma, one had leiomyosarcoma, and one had desmoplastic small round cell tumor. In the group of five patients with ES, the response evaluation showed SD in two patients, while three patients experienced disease progression. The DCR was 40% [34]. In a study that enrolled 83 patients to evaluate the efficacy of TC in recurrent and refractory pediatric solid tumors, 17 patients were diagnosed with ES. Among them, six patients exhibited a CR or PR, resulting in an ORR of 35.2%. Regarding toxicity, the patients received a total of 307 courses of TC. Of these, 163 (53%) courses were associated with grade 3 or 4 neutropenia, 84 (27%) courses with grade 3 or 4 anemia, and 136 (44%) courses with grade 3 or 4 thrombocytopenia [35]. In our study, the ORR was 10.6%, the DCR was 30.6%, the mPFS was 3.06 months, and the mOS was 6.16 months. Any hematological side effects were observed in 55 (80.8%) patients, with grade 3–4 hematological side effects observed in 35 (51.4%) patients. Grade 3–4 anemia was observed in 28 (41.1%) patients, and grade 3–4 thrombocytopenia was observed in 18 (26.4) patients. Erythrocyte transfusion was required in 20 (29.4%) patients, and platelet transfusion was required in 10 (14.7%) patients. Despite the occurrence of grade 3–4 neutropenia in 25 (36.7%) patients, only 5 (7.3%) developed febrile neutropenia, and no treatment-related deaths were observed.

Most studies in the literature focus on pediatric populations. Only a few studies have addressed metastatic ES in adults, and these studies typically used small sample sizes. Therefore, there are very few studies available for direct comparison in the adult population. The ORR in our study was lower than that in pediatric studies, which we attribute to 46.6% of patients being metastatic at the time of diagnosis and half of the patients receiving TC as a third-line treatment or later.

Due to the nature of this study, several limitations exist. Potential bias may have been introduced due to its retrospective design. Additionally, because this was a multicenter study, evaluations were not performed by the same radiologist, and adverse effects may have been underreported. We acknowledge that one of the limitations of this study is the lack of uniformity in the initial treatment and the absence of genetic fusion data in the patients.

## 5. Conclusions

To the best of our knowledge, this study is the largest real-world data investigation into the efficacy and safety of the TC regimen in adult patients with metastatic ES. No standard treatment options exist for second-line and subsequent lines of therapy. This study showed that the use of the TC regimen in the second line resulted in better efficacy and overall survival outcomes compared to its use in the third line or later. However, in the entire population, the TC regimen demonstrated only a modest effect on metastatic ES. TC can be considered one of the palliative treatment options for metastatic ES, but this disease remains a significant unresolved challenge.

## Figures and Tables

**Figure 1 cancers-17-00550-f001:**
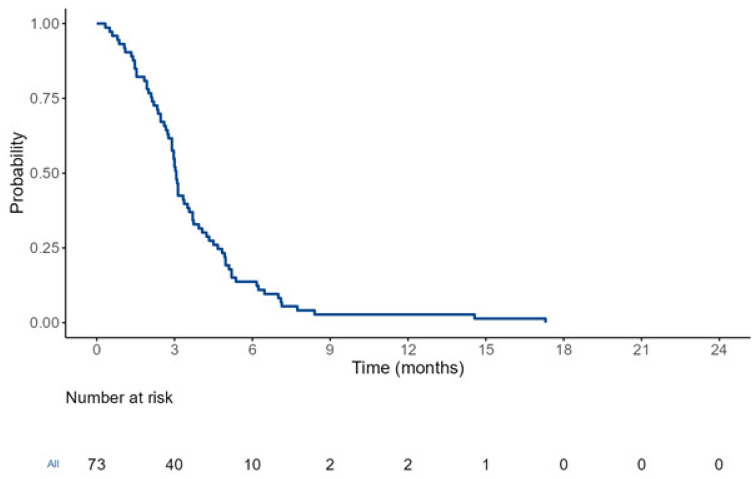
Kaplan–Meier plot of topotecan–cyclophosphamide treatment. Median PFS was 3.06 months (95% CI, 2.91–3.22).

**Figure 2 cancers-17-00550-f002:**
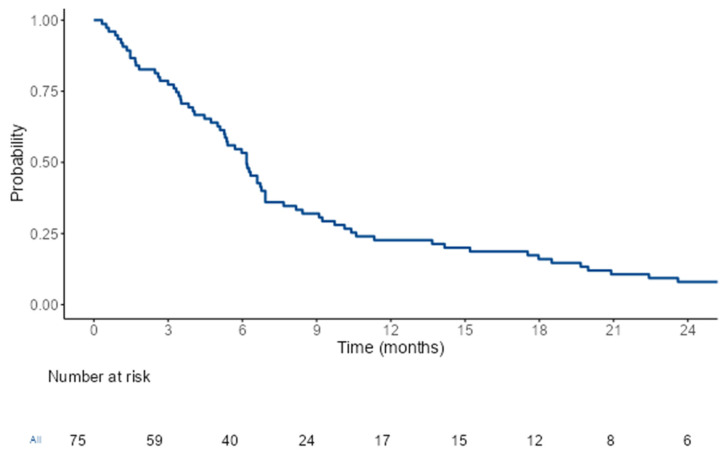
Kaplan–Meier plot of topotecan–cyclophosphamide treatment. Median OS was 6.16 months (95% CI, 5.14–7.18).

**Figure 3 cancers-17-00550-f003:**
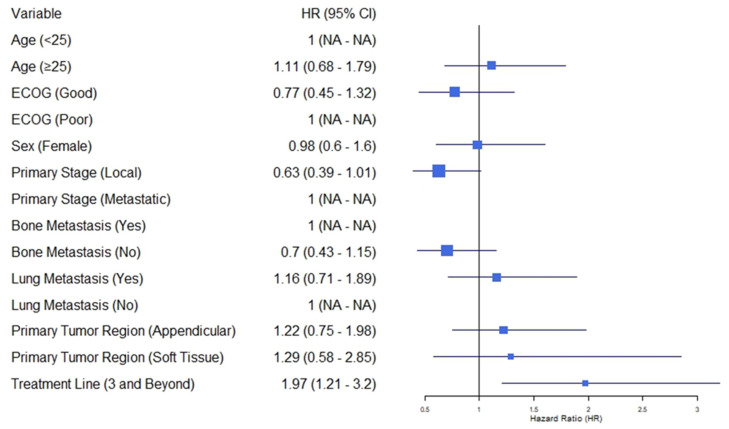
Forest plot for univariate Cox proportional hazards regression analyse. The horizontal bars in the forest plot represent the 95% confidence intervals for the hazard ratios. Variables where the confidence interval crosses the vertical line at HR = 1 were not statistically significant. Abbreviations: HR: hazard ratio; CI: confidence interval; ECOG PS: Eastern Cooperative Oncology Group Performance Status.

**Table 1 cancers-17-00550-t001:** Patients characteristics (n = 75).

Variable	Value
Patients, n (%)	75 (100)
Age (years), median	23 (18–52)
Sex, n (%) Male Female	50 (66.6) 25 (33.3)
ECOG PS, n (%) PS: 0–1 PS: 2	54 (72) 21 (28)
Primary Tumor Site, n (%) Axial Appendicular Soft tissue	35 (46.6) 32 (42.6) 8 (10.6)
Patients by Tumor Stage at Initial Diagnosis, n (%) Localized De novo metastatic	40 (53.3) 35 (46.6)
Metastatic sites Lung Bone Lymph node Liver Others *	37 (49.3) 34 (45.3) 15 (20) 5 (6.6) 4 (5.3)
Primary surgery Yes No	50 (66.6) 25 (33.3)
TC regimen was administered Second-line Third-line Fourth-line or later	38 (50.6) 22 (29.3) 15 (20)
Radiotherapy before TC regimen Primary sites Palliative	42 (56) 18 (24)
Chemotherapy before TC regimen IE/VDC GD VIT ICE HDT/HCT	75 (100) 28 (37.3) 22 (29.3) 18 (24) 2 (2.6)
Chemotherapy after TC regimen Yes No	27 (36) 48 (64)

All data are presented as the median (range) or absolute number (%). Abbreviations: ECOG PS—Eastern Cooperative Oncology Group Performance Status; IE/VDC—ifosfamide and etoposide/vincristine, doxorubicin, and cyclophosphamide; GD—Gemcitabine–docetaxel; VIT—vincristine–irinotecan–temozolomide; ICE—ifosfamide–carboplatin–etoposide; HDT/HCT—high-dose therapy followed by hematopoietic cell transplant. * Kidney: n  =  2, central nervous system: n  =  1, adrenal gland: n = 1.

**Table 2 cancers-17-00550-t002:** Univariate analysis of baseline characteristics for overall survival.

		N (%)	HR	CI	*p*
Variables					
Age	<25	49 (65.3)	Ref.		
	≥25	26 (34.7)	1.11	0.68–1.79	0.679
ECOG	Poor (2)	21 (28)	Ref.		
	Good (0–1)	54 (72)	0.77	0.45–1.32	0.346
Sex	Male	50 (67.6)			
	Female	25 (33.3)	0.98	0.60–1.60	0.945
Primary Stage	Metastatic	35 (46.7)	Ref.		
	Local	40 (53.3)	0.63	0.39–1.01	0.054
Metastatic Region	Bone ^Yes^	34 (45.3)	Ref.		
	Bone ^No^	41 (54.7)	0.70	0.43–1.15	0.159
	Lung ^Yes^	37 (49.3)	Ref.		
	Lung ^No^	38 (50.7)	1.16	0.71–1.89	0.434
Primary Tumor Region	Axial	35 (45.7)	Ref.		
	Appendicular	32 (42.7)	1.22	0.75–1.98	0.434
	Soft Tissue	8 (10.7)	1.29	0.58–2.85	0.536
Treatment Line	2	38 (50.7)	Ref.		
	3 and Beyond	37 (49.3)	1.97	1.21–3.20	0.006

Variables were analyzed using univariate Cox proportional hazards regression to determine their association with overall survival. Abbreviations: HR: hazard ratio; CI: confidence interval; ECOG PS: Eastern Cooperative Oncology Group Performance Status. “^Yes^” indicates the presence of bone or lung metastasis, while “^No^” indicates the absence of bone or lung metastasis.

**Table 3 cancers-17-00550-t003:** Best response during topotecan–cyclophosphamide treatment.

Outcome for Second Line	No (%)
Complete response	0 (0)
Partial response	5 (13.2)
Stable disease	14 (36.7)
Objective response rate	5 (13.2)
Disease control rate	19 (50)
Outcome for third line or later	
Complete response	0 (0)
Partial response	3 (8.1)
Stable disease	1 (2.7)
Objective response rate	3 (8.1)
Disease control rate	4 (10.8)
Outcome for entire group	
Complete response	0 (0)
Partial response	8 (10.7)
Stable disease	15 (20)
Objective response rate	8 (10.7)
Disease control rate	23 (30.7)
Progressive disease	39 (52)
Not Evaluated *	13 (17.3)

* Radiological evaluation could not be performed for 10 patients due to death and for 3 patients due to discontinuation of topotecan–cyclophosphamide regimen because of adverse effects. Treatment-related responses were interpreted according to the Response Evaluation Criteria in Solid Tumors 1.1 criteria.

**Table 4 cancers-17-00550-t004:** Topotecan–cyclophosphamide regimen-related adverse events according to the Common Terminology Criteria for Adverse Events v 5.0.

Toxicity	N (%)
Any hematologic toxicity	55 (80.8)
Grade 3–4 Hematologic toxicity	35 (51.4)
Grade 3–4 anemia	28 (41.1)
Grade 3–4 thrombocytopenia	18 (26.4)
Grade 3–4 neutropenia	25 (36.7)
Febrile neutropenia	5 (7.3)
Patient requiring transfusion Erythrocytes Platelets	20 (29.4) 10 (14.7)
Not Evaluated *	7 (9.3)

* Due to early deaths following the first cycle in seven patients, the evaluation of side effects could not be performed.

## Data Availability

Data will be available from the corresponding author upon reasonable request.

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
