# Peer review of "Efficacy and Safety of the Topotecan–Cyclophosphamide Regimen in Adult Metastatic Ewing Sarcoma: A Large, Multicenter, Real-World Study"

_cancers, 2025, doi:10.3390/cancers17030550_

Round 1
Reviewer 1 Report
Comments and Suggestions for Authors
Metastatic Ewing sarcoma (ES)has a very dismal prognosis. As well cited by the Authors good results can be obtained only in NON metastatic ES limited to the bone in appendicular area of the skeleton and mainly in children and adolescents. In this specific population 70% of 3 years free survival can be recorded.
On the contrary metastatic ES heavily pretreated in first line therapy , mainly if the primary sarcoma was in extraskeletal area and the Patients are adults the median survival at 3 years is less than 10% in extraordinary selected population. Otherwise the survival is almost zero.
The present paper is a retrospective, multiinstitutional report of the activity of the chemo combination between Topotecan and Cyclophosphamide in adult people with metastaic ES.
Many different combination of drugs have been evaluated as second line therapy: TEMIRI, ICE, TP, High dose chemotherapy with bone marrow transplant.
All these attempt offered poor results in term of PFS and OS.
The TC combination utilized in this daily practice protocol, according with the Authors offered a slightly advantage in partial response , but the median progression-free survival of 3.06 months (95% CI, 2.91–3.22), 37 and the median overall survival of 6.16 months (95% CI, 5.14–7.18) with no surviving Patients at the follow up cannot be reported as a great result.
On the other 55% of grade 3-4 myelosuppression for a totally palliative treatment cannot be considered as a "manageable toxicity."
In conclusion I suggest a more thoughtful conclusion by the Authors . TC can be considered one of the palliative options in metastatic ES but this disease remains a great unsolved problem.
Author Response
Comments 1: [Given the PFS and OS results, as well as the observation of side effects in 55% of patients, should the conclusion be reconsidered to reflect these findings more carefully]
Response 1: [In our study, the median progression-free survival was 3.06 months (95% CI, 2.91–3.22) and the median overall survival was 6.16 months (95% CI, 5.14–7.18). Patients who received the TC regimen in the second line showed longer overall survival (7.55 months 95% CI, 5.37–14.17) compared to those who received it in the third line or later (5.70 months 95% CI, 4.07–6.60) (p=0.005). When the TC regimen was used in the second line, the disease control rate was 50%, while in the third line or later, the disease control rate was 10.8%. In the entire group, the disease control rate was 30.7%. This study showed that the use of the TC regimen in the second line resulted in better efficacy and overall survival outcomes compared to its use in the third line or later. However, in the entire population, the TC regimen demonstrated only a limited effect on metastatic ES. TC can be considered one of the palliative treatment options for metastatic ES, but this disease still remains an unresolved problem.]
First of all, we would like to express our sincere thanks for your valuable comments and contributions. Although our study showed that the use of the TC regimen in the second line results in better disease control rate and overall survival compared to the third line and beyond, we acknowledge that, when considering the entire group, the effect of the TC regimen on metastatic ES is limited. Additionally, while the 55% incidence of grade 3-4 side effects is manageable, it is still a relatively high rate. Taking all these factors into account, we have concluded that the TC regimen should be considered one of the palliative treatment options for metastatic ES.
Kind regards.
Reviewer 2 Report
Comments and Suggestions for Authors
Dear Authors
1. keywords -you should add - 2-nd line chemotherapy, it should be: Ewing sarcoma, 2-nd line chemotherapy, Topotecan, Cyclophosphamide,...
2. Introduction:
Second line treatment regimens - it is necessary to add doses of cytostatics, lines 69-73
3. Patients and data collections -
do you use iv fluids to cyclophosphamide
is in Turkey one, the same for every one a protocol for 2nd line chemotherapy
how many cycles of TC did the patients receive?
4. Dscussion - what do you know about antiGD2 in treatment of ESA?
Author Response
Comments 1: [The addition of second-line chemotherapy to the keywords.]
Response 1: [second-line chemotherapy has been added to the keywords.]
Comments 2: [Introduction: Second-line treatment regimens - the addition of cytostatic doses is required.]
Response 2: [Introduction: Cytostatic doses have been added to second-line treatment regimens.]
Comments 3:[ Do you use IV fluids for cyclophosphamide?]
Response 3: [IV hydration support has been used for uroprotection during cyclophosphamide administration.]
Comments 4: [is in Turkey one, the same for every one a protocol for 2nd line chemotherapy?]
Response 4: [In Turkey, there is no standardized second-line treatment protocol for metastatic ES, and different treatment protocols are applied across clinics.]
Comments 5: [how many cycles of TC did the patients receive?]
Response 5: [The median number of cycles was four (range: 1–12)]
Comments 6: [ Discussion - what do you know about antiGD2 in treatment of ESA?]
Response 6: [Gangliosides are molecules composed of glycosphingolipids associated with one or more sialic acid residues. Disialoganglioside GD2, a glycosphingolipid that plays a role in cell proliferation, differentiation, survival, and tumor progression, is a tumor-associated antigen expressed in various malignancies, including neuroblastoma, but is found at limited levels in normal tissues. Immunotherapy using GD2-targeting antibodies is the standard first-line treatment for high-risk neuroblastoma. GD2 expression is also observed in other pediatric solid tumors, such as ES, osteosarcoma, and others. In ES, GD2 expression levels range from 40% to 90% in biopsy samples, making it a potentially valuable therapeutic target in this malignancy. The addition of the anti-GD2 antibody dinutuximab beta to standard chemotherapy regimens has been evaluated for its efficacy in three patients with metastatic, GD2-positive ES or Ewing-like sarcoma. The treatment was well-tolerated, and all patients achieved complete remission without evidence of relapse. First-line anti-GD2 immunotherapy in patients with metastatic, GD2-positive ES or Ewing-like sarcoma represents a promising therapeutic option that warrants further clinical evaluation.]
First of all, I would like to thank you for your valuable feedback and contributions. Based on your suggestions, specific changes have been made in the manuscript, and a paragraph related to anti-GD2 has been added.
Kind regards.
Reviewer 3 Report
Comments and Suggestions for Authors
The authors retrospectively reported on the efficacy and safety of TC therapy in adult recurrent/metastatic Ewing's sarcoma patients. Although this is important information, the following modifications are recommended to improve the quality of the paper:
1. Table 1 summarizes the characteristics of the patients, but a breakdown of the fusion gene should also be included ("EWS-FLI1", "other fusion genes", and "not tested" should be included at a minimum). If detailed information on pathological diagnosis is not available, this should be stated as a Limitation of the paper.
2. Although prognostic factors related to OS are evaluated in Table 2, it would be preferable to present the hazard ratios by factor and their confidence intervals in a forest plot as a figure rather than OS for each item.
3. L270-281 mentions the Phase III (rEECur) study, which is necessary information for the discussion of this paper. However, this study was conducted with an adaptive design, in which four groups (HDIFO, IT, TC, and GD) were initially compared, then the GD group dropped out first, followed by the IT group in the Phase II part, and finally HDIFO and TC were compared, making it impossible to simply compare the results of all included patients in each group (The results of TC and IDIFO mentioned in the text are appropriate in themselves). I therefore request that a brief description of the design and evaluation steps of this Phase III study be added.
Author Response
Comments 1: [Table 1 summarizes the characteristics of the patients, but a breakdown of the fusion gene should also be included ("EWS-FLI1", "other fusion genes", and "not tested" should be included at a minimum). If detailed information on pathological diagnosis is not available, this should be stated as a Limitation of the paper.]
Response 1: [Since our study includes patients from the years 2011 to 2020, fusion genes could not be analyzed in the patients' pathology. We acknowledge that this is a limitation of our study.]
Comments 2: [Although prognostic factors related to OS are evaluated in Table 2, it would be preferable to present the hazard ratios by factor and their confidence intervals in a forest plot as a figure rather than OS for each item]
Response 2: [The table has been revised according to your suggestions.]
Comments 3:[ L270-281 mentions the Phase III (rEECur) study, which is necessary information for the discussion of this paper. However, this study was conducted with an adaptive design, in which four groups (HDIFO, IT, TC, and GD) were initially compared, then the GD group dropped out first, followed by the IT group in the Phase II part, and finally HDIFO and TC were compared, making it impossible to simply compare the results of all included patients in each group (The results of TC and IDIFO mentioned in the text are appropriate in themselves). I therefore request that a brief description of the design and evaluation steps of this Phase III study be added.]
Response 3: [Detailed information about the design of the REECUR study has been added to the manuscript according to your suggestions]
First of all, we would like to thank you for your feedback and valuable contributions. We acknowledge that the absence of fusion gene analysis is a limitation of our study. We have provided more details about the REECUR study in our manuscript, and your contributions have played an important role in making our study more comprehensive.
Kind regards.
Reviewer 4 Report
Comments and Suggestions for Authors
The manuscript by Tunbekici et al describes a retrospective cohort of adult patients with relapsed metastatic Ewing sarcoma who received cyclophosphamide-topotecan as second (or more) line of therapy. This is an interesting report and adds to the literature.
There are however several points that the authors should address prior to publication:
1. In modern day therapy for Ewing sarcoma, fusion status is important in distinguishing Ewing sarcoma from other small round blue cell tumors. Please clarify whether molecular confirmation was done or could be achieved. Analysis limited only to molecularly confirmed Ewing sarcoma would be helpful
2. Can the authors comment on the effect of cyclophosphamide-topotecan as second line versus third+ line of therapy? What was the efficacy for the 38/75 patients for whom cyclophosphamide-topotecan was second line therapy? It might be helpful to break down the survival by lines of therapy to see if there might be an effect.
3. Please acknowledge that one of the limitations of this retrospective study is the lack of uniformity in initial therapy.
Author Response
Comments 1: [In modern day therapy for Ewing sarcoma, fusion status is important in distinguishing Ewing sarcoma from other small round blue cell tumors. Please clarify whether molecular confirmation was done or could be achieved. Analysis limited only to molecularly confirmed Ewing sarcoma would be helpful.]
Response 1: [Small blue round cell sarcomas (SBRCS) are a heterogeneous group of mesodermal tumors. Ewing sarcoma (ES) is the most common type of SBRCS. The overlapping histological and morphological features make it challenging to classify SBRCS into specific subgroups based solely on morphology. Advances in cytogenetics and molecular genetics have made it easier to detect tumor-specific alterations in certain subgroups of SBRCS. In our study, due to the inclusion of patients between 2011 and 2020, fusion tests were not performed on the patients. In Turkey, NGS or FISH testing for EWSR1-FLI1 or other fusion tests has only been implemented in the last 3-4 years. However, in our study, patients showing intense and membranous CD99 staining pattern in neoplastic cells by immunohistochemistry, and where other SBRCS types were excluded, were included. Cases with diagnostic uncertainty were excluded from the study. Nevertheless, we acknowledge that the lack of fusion testing is an important limitation of our study.]
Comments 2: [ Can the authors comment on the effect of cyclophosphamide-topotecan as second line versus third+ line of therapy? What was the efficacy for the 38/75 patients for whom cyclophosphamide-topotecan was second line therapy? It might be helpful to break down the survival by lines of therapy to see if there might be an effect.]
Response 2: [Patients who received the TC regimen in the second line demonstrated longer overall survival (7.55 months, 95% CI, 5.37–14.17) compared to those who received it in the third line or later (5.70 months, 95% CI, 4.07–6.60) (p=0.005). When the TC regimen was used in the second line, the disease control rate was 50%, whereas in the third line or later, the disease control rate was 10.8%.]
Comments 3:[ Please acknowledge that one of the limitations of this retrospective study is the lack of uniformity in initial therapy.]
Response 3: [We acknowledge that one of the limitations of our retrospective study is the lack of uniformity in initial therapy.]
First of all, we would like to thank you for your valuable feedback and contributions. We acknowledge that the lack of fusion gene testing is a limitation of our study. Additionally, we have highlighted in the manuscript that the use of the TC regimen in the second line resulted in longer OS and better DCR. Your contributions have played an important role in making our manuscript more comprehensive.
Kind regards.
Round 2
Reviewer 3 Report
Comments and Suggestions for Authors
The authors made appropriate revisions to the paper based on my suggestions. I consider the revised paper worthy of publication.